# Detection of specific uncultured bacteriophages by fluorescence *in situ* hybridisation in pig microbiome

**Line Jensen Ostenfeld, Patrick Munk, Frank M. Aarestrup, Saria Otani** *

Research group for Genomic Epidemiology, National Food Institute, Copenhagen, Denmark

* saot@food.dtu.dk

## Abstract

Microbial communities have huge impacts on their ecosystems and local environments spanning from marine and soil communities to the mammalian gut. Bacteriophages (phages) are important drivers of population control and diversity in the community, but our understanding of complex microbial communities is halted by biased detection techniques. Metagenomics have provided a method of novel phage discovery independent of *in vitro* culturing techniques and have revealed a large proportion of understudied phages. Here, five jumbophage genomes, that were previously assembled *in silico* from pig faecal metagenomes, are detected and observed directly in their natural environment using a modified phageFISH approach, and combined with methods to decrease bias against large-sized phages (*e.g.*, jumbophages). These phages are uncultured with unknown hosts. The specific phages were detected by PCR and fluorescent *in situ* hybridisation in their original faecal samples as well as across other faecal samples. Co-localisation of bacterial signals and phage signals allowed detection of the different stages of phage life cycle. All phages displayed examples of early infection, advanced infection, burst, and free phages. To our knowledge, this is the first detection of jumbophages in faeces, which were investigated independently of culture, host identification, and size, and based solely on the genome sequence. This approach opens up opportunities for characterisation of novel *in silico* phages *in vivo* from a broad range of gut microbiomes.

## Introduction

Viruses of bacteria (phages) are an important part of microbial communities, including human and animal gut microbiomes [1,2]. The gut microbiome has been shown to affect the host in many aspects from nutrient uptake [3,4] to diabetes [5], obesity [6], and neuropsychiatric disorders [7,8]. Despite the large proportion and diversity of phages in the gut microbiome, the impact of phages on the bacterial community remains poorly understood as bacterial species have been the main focus of microbiome studies.

Different novel phage types have been detected as new detection methods are introduced, highlighting bias in previous research methods [9,10]. These include for example

**Data Availability Statement:** All data are in the Supporting Information file.

**Funding:** Yes - This work was mainly funded by the Novo Nordisk Foundation Grant NNF16OC0021856

to F.M.A. The funders had no role in study design, data collection and analysis, decision to publish, or preparation of the manuscript.

**Competing interests:** The authors have declared that no competing interests exist.

jumbophages, aggregating phages (phages that tend to cluster [10,11]), and phages that infect unculturable bacteria. Therefore, it is important to be able to study all types of phages in a community, in order to properly understand their role and impact on the microbial ecosystem. This too, requires novel methods.

Metagenomic analyses of complex microbial communities have emerged over the last two decades revealing a large proportion of microbes that were previously undetected, including bacteria, viruses, and archaea [12–15]. Metagenomics are less biased against phages of different morphologies and infection-patterns, as culturing is not required. The possibilities of comparative genomics and genome assemblies from metagenomic data have thus led to an increase in novel species identification of bacteria, archaea, and phages. A lot can be learned from the genome sequence of novel species, but phenotypic characterisation and gene functions are still largely culture-dependent. However, the road from *in silico* genome to *in vitro* isolation is long and complicated [16–18].

Few phages have been isolated based on metagenomics sequencing data. Phages are dependent on the availability of a specific bacterial host for replication, as well as several other biological factors that are needed for their survival. Their genomes, however, are extremely diverse and leave little information about hosts, optimal growth conditions, and alternate life cycles. Over the past decade, a series of webserver tools for phage-host prediction have emerged based on sequence similarity, CRISPR spacer matches, and similarity to known phage-host pairs [19], including; PHERI, HostPhinder, Host Taxon Predictor, and WIsH [20–23]. These tools can provide valuable methods for narrowing down potential host species, but their efficiency is limited to well-studied phage-host pairs and well-characterised host species. As a majority of microbial species are yet unknown [24], and phage culturing is also subject to environmental conditions [10,11], *in vitro* isolation of phages that are discovered in metagenomics presents a great challenge.

A successful *in vitro* isolation of a metagenomics-derived phage is the CrAss-phage [25]. CrAss phage genome was discovered in 2014 in human gut microbiome sequences, and it was shown to occupy up to 90% of the human gut viral community [25]. Despite the abundance of this novel phage, it was not until 2018 a member of the CrAss-phage family was first isolated [26] and to this date only four members of the CrAss phage family have been isolated [26–28].

As more phages are discovered *in silico*, it has become more apparent that traditional methods of phage isolation are biased towards smaller phages with known life cycle traits [10]. However, phages with genomes of >200,000bp (jumbophages) are also found in metagenome studies despite the underrepresentation of sequenced and isolated phages of this size [9,29,30]. Phage genomes are packed tightly to maximise genetic storage in the small capsids. Phages with larger than average genomes (around 150-175kb [31]) are thus likely to have a larger capsid, which might alter the behaviour of some of these phages *in vitro*. Traditional isolation methods, such as filtering and normal plaque assays (0.7% agar), will not be suitable for jumbophage isolations without modification, as diffusion of the large particles is inhibited by the dense agar matrix [10]. As organisms isolated *in silico* have proven difficult to isolate, the challenge lies in studying and characterising important phages in a culture-independent method. Several approaches have been suggested including microfluidic PCR [32], viral tagging [33] and phageFISH [34].

All three methods can aid in phage host identification independently of culture, and can potentially answer questions about the host range of uncultured phages. However, microfluidic PCR and viral tagging do not allow characterisation of viral behaviour or visual detection.

PhageFISH can potentially characterise phages without the need for culturing [34]. Phage-FISH utilises the known phage genome to produce several long DNA probes that are specific to each phage for fluorescent detection and observation along with bacterial probes. The

method was developed for improved characterisation of phage-host interactions in marine systems. Allers et al. were able to detect intra- and extracellular phage DNA, as well as perform host identification and quantification. It was also possible to study different waves of infection and discriminate between early and late infection stage [34]. As the method uses specific phage probes designed from the phage genome, phageFISH could potentially be employed for the detection of uncultured phages discovered *in silico* in metagenomics samples, including faecal microbiomes.

In this study, we employed the phageFISH method [34] to visualise five jumbophages *in silico* that were previously assembled from Danish pig faecal metagenomes (used in Al-Shayeb et al., 2020) thereby confirming that the assembled genomes are derived from phages and are not falsely assigned as phages based on the metagenomic data. PhageFISH allows visualisation directly in the native faecal matrix via specific DNA probes. To our knowledge, this is the first time phageFISH has been used to detect and visualise previously uncultured jumbophages directly in faecal samples, linking genomes of phages discovered *in silico* to *in vitro* visualisation.

## Material and methods

### Sample collection

Pig faecal samples were collected as part of the Danish surveillance project VetForligII from a number of farms in Denmark between 2014 and 2016. All faecal samples were collected, handled and stored similarly. Faeces were stored at -80˚C until use.

Five novel jumbophage genomes were assembled based on sequencing data from 10 pig faecal metagenomes from the above study. The jumbophage genomes are published in Al-Shayeb et al. 2020 as part of a large study to identify novel jumbophages [9]. Each phage genome included here was based on similar sequence assemblies from two different faecal samples. The novel phages are denoted Phage A, B, C, D, or E (Table 1, S1 Table in S2 File) (genome accession numbers as published by Al-Shayeb et al., 2020: ERS4026424—ERS4026427—ERS4026420 —ERS4026428—ERS4026423—ERS4026426—ERS4026430—ERS4026425—ERS4026421— ERS4026431—ERS4026432—ERS4026422—ERS4026429).

### Isolation of large-sized viral population from faeces

Large-sized viral population was obtained from the faecal sample as described previously by Saad et al. 2019 with modifications. One gram frozen faecal matter was suspended thoroughly in 40ml SM-buffer (0.1M NaCl, 10mM MgSO$_4$, 50mM Tris-HCl pH 7.5, 0.01% (w/v) gelatin). The suspension was shaken at 800rpm overnight at 4˚C to maximise dissociation of phages. The suspension was then centrifuged at 5,000x*g* for 10 minutes to pellet large bacteria and debris. The supernatant was then centrifuged at 15,000x*g* for 1 hour at 4˚C to pellet large phages. The pelleted phages were resuspended in 5ml SM-buffer, and the centrifugation process was repeated before adding an equal volume of chloroform to eliminate bacteria. The final viral population was centrifuged at 15,000x*g* again and suspended in 2ml SM-buffer.

### Large viable phage detection with plaque assay

Standard plaque assays were adapted for jumbophage isolation as described by Saad et al. 2019 with few modifications. Two strains of *Escherichia coli*: *E.coli* 11303 and *E. coli* BAA-1025, were grown on lysogeny broth (LB) agar plates overnight and 3–4 colonies were inoculated into LB broth and incubated for 2–3 hours to reach exponential growth phase. *E. coli* strains were used as potential hosts for the jumbophages in order to be able to isolate them on plates

**Table 1. Detection of assembled phage genomes in extracted viral population from each faecal sample by PCR amplification.**

| | | Primer pairs used for probe detection | | | | | | | | | | | | | |
|---|---|---|---|---|---|---|---|---|---|---|---|---|---|---|---|
| | | A5 | A6 | B3 | B4 | B7 | C5 | D3 | D6 | E1 | E2 | E3 | E4 | E5 | E6 | E7 |
| **Phage groups in their faecal samples** | | | | | | | | | | | | | | | | |
| **Group A** | F12 | + | + | - | - | - | (+) | - | - | - | - | - | - | - | - | - |
| | F33 | + | + | (+) | - | - | + | + | - | - | - | - | - | - | - | - |
| **Group B** | F42 | + | + | + | + | + | + | + | - | - | - | - | - | - | - | - |
| | F71 | + | + | + | + | + | + | + | + | - | - | - | - | - | - | - |
| **Group C** | F67 | - | - | (+) | - | - | + | - | - | - | - | - | - | - | - | - |
| | F78 | + | + | - | - | - | + | - | - | - | - | - | - | - | - | - |
| **Group D** | F95 | + | + | (+) | - | - | - | + | + | - | - | - | - | - | - | - |
| | F101 | + | + | + | + | + | - | + | + | + | + | + | + | + | + | - |
| **Group E** | F49 | + | + | - | - | - | - | - | + | + | + | + | + | + | + | - |
| | F59 | - | - | - | - | - | - | - | ND | + | + | + | + | + | + | + |

[+]Single band at 300bp.

[(+)]Faint band at 300bp.

[-]No band, smudged bands, or inconclusive data.

[ND]No data.

to detect their large size only (see Results section). 250μl of the exponential phase bacteria and 100μl viral population were co-incubated for 10 minutes before adding 5ml LB broth (or BHI broth) with 0.35% agar supplemented with 10mM $CaCl_2$ and 10mM $MgSO_4$. The bacteria-phage mix was immediately poured on LB (or BHI) agar plates. All plates were inoculated in 4 replicates and incubated at 30°C or 37°C as well as aerobically or anaerobically, for 24–48 hours.

Isolation of jumbophages *in vitro* was attempted on other hosts than *E. coli* strains as above. This was done to increase the success in isolating jumbophages from the rich faecal samples. The same above-mentioned phage assays were used with the other bacterial taxa (details in S2 File). Phages of different size (3kb– 170kb) could be isolated from *S. aureus*, *B. cereus*, *S. marcescens*, *S. sonnei*, and all strains of *E. coli*.

## DNA extraction

DNA was extracted from each faecal large viral population using Blood and Tissue kit (QIA-GEN, Denmark). The large-sized viral population sample, isolated as described above (Isolation of large-sized viral population from faeces), was centrifuged at 15,000x*g* for 1 hour. Pellet was resuspended in 200μl ATL buffer and 20μl proteinase K, and lysed overnight at 56°C with shaking. Protocol "Purification of Total DNA from Animal Tissues" was followed afterwards according to the manufacturer's instructions. Samples eluted in 2x50μl preheated AE buffer and DNA concentrations were measured with Qubit (Invitrogen).

## Large phage detection with sequencing

DNA yields from the faecal viral population were subjected to Illumina sequencing to validate the phages are of large size. The DNA was used to build the libraries using Nextera XT DNA library preparation kit (Illumina) following manufacturer's instruction, and sequenced on Illumina NextSeq 550 platform. Illumina sequencing data were quality and adapter trimmed

using bbduk from the bbmap suite (https://sourceforge.net/projects/bbmap/; v38.23) using the following settings: qin = auto, k = 19, rref = adapters.txt, mink = 11, qtrim = r, trimq = 20, minlength = 50, tbo, ziplevel = 6, overwrite = t, and statscolumns = 5. To evaluate the phage assembly length to be long, the quality reads were *de novo* assembled in Geneious using standard settings.

## Primer design and probe synthesis

Several DNA primers amplifying a 300bp fragment (S2 Table in S2 File) of the targeted phage genomes were designed for each phage (S3 Table in S2 File). Briefly, for each phage target genome from Al-Shayeb et al, at least one other high-similarity genome was identified and aligned against the target genome using the progressive Mauve algorithm as implemented in Geneious. The resulting alignments were used to visually identify the most conserved regions (>300bp) which were extracted. In these high-identity regions we used Geneious built-in implementation of Primer3 (v.2.3.7) to generate primer pairs using the default parameters, except for length to be: 20–30, optimal primer length: 25, Tm: 57–63 and product length: 300 bp.

DNA probes were synthesised using the designed primers followed by PCR DIG-probe Synthesis kit (Roche) according to the manufacturer's instructions in 50μl reaction volumes. A touchdown reaction cycle was used for PCR product amplification (94°C 4min, 35 cycles of [94°C 30s, 60°C 30s (-0.20°C/cycle), 72°C 30s], 72°C 4min).

45μl PCR product was purified with Gene Clean Turbo for PCR kit (MP Biomedicals) according to the manufacturer's instructions. Probes were eluted in 30μl sterile nuclease-free water. The PCR product size, concentration, and integrity were evaluated on Bioanalyzer 2100 (Agilent).

A single unlabelled probe from each sample was sequenced with Sanger sequencing to confirm the probe sequence (Eurofins).

## PhageFISH

The protocol described in this peer-reviewed article is published on protocols.io ([https://dx.doi.org/10.17504/protocols.io.rm7vzb7z2vx1/v1], [dx.doi.org/10.17504/protocols.io.4r3l273wqg1y/v1], [dx.doi.org/10.17504/protocols.io.kqdg3931pg25/v1], [dx.doi.org/10.17504/protocols.io.dm6gpjop8gzp/v1] and are included for printing purposes as S1 File. Prior to performing the fluorescence in situ hybridisations, the targeted phages were detected across all faecal samples included in this study (Table 1) by PCR amplification using the same primers that were also designed for probe synthesis with the same conditions described above. Fluorescence *in situ* hybridisation of phages was adapted from Barrero-Canosa and Moraru 2019 [35]. All buffers are listed in S4 Table in S2 File. Briefly, a 10μl loop-full of frozen faecal matter was suspended in 50μl PBS. 10μl of the faecal suspension were placed directly on a poly-L-lysine coated glass slide and smeared into a thin layer. The smeared sample was dried for 30 minutes at room temperature before fixing in 500μl of 1% paraformaldehyde (v/v) for 1 hour at room temperature. Slides were rinsed in PBS and moved to a permeabilisation buffer (S4 Table in S2 File) for 1 hour on ice. The slides were gently rinsed in PBS and sterile water before incubation in 10mM HCl for 10 minutes, followed by a final rinse in PBS and sterile water, and dried in 96% ethanol.

Cyanine-labelled bacterial 16S rRNA probe, EUB338, and negative control, nonEUB338 (TAGCopenhagen, S2 Table in S2 File), were hybridised by incubation in hybridisation buffer I for 3 hours at 46°C. After hybridisation, the slides were incubated in wash buffer I at 48°C (S4 Table in S2 File).

Phage probes were hybridised to the glass slides according to Barrero-Canosa & Moraru 2019 (SI methods) and stored at -20°C.

For mounting and DAPI DNA-universal staining, the slides were thawed for 10 minutes at room temperature and 10μl SlowFade Gold with 5μg/ml DAPI were added to the slides and covered by 24x50mm cover glass. Samples were sealed with clear nail polish and visualised with fluorescence microscope (Olympus BX53, PlanApo N 60x objective). PhageFISH images were captured using CellSense software (Olympus) and processed in ImageJ.

### Negative controls

The signal from the nonsense bacterial probe nonEUB338-Cy5 (Fig 1, S2 Table in S2 File) further discriminates between phage signals and background signals as these overlaps well with background fluorescence during image processing (Fig 1, S5a, S5b and S6 Figs in S2 File), *i.e.*, the nonsense bacterial probe nonEUB338-Cy5 can show non-targeted fluorescence as it binds to the faecal debris. Despite prevalent auto-fluorescence in the faecal samples due to the high and natural debris content, which explains the auto-fluorescence that appears from the nonsense bacterial probe nonEUB338-Cy5, phage signals can be clearly distinguished (Fig 1), and thus the phage signal amplification in the FITC filter spectrum was accepted in the downstream analyses and detection of phage signals.

## Results

### Isolation of large viral population

The population of large phages was isolated from each faecal sample for DNA purification and probe synthesis. The efficiency of the method was evaluated based on phage recovery by plating the large viral population on common phage isolation hosts, *e.g.*, *E. coli* 11303 and *E. coli* BAA-1025 in dilute LB agar to support plaque formation from large phages. All tested large viral populations were able to produce plaques on the *E. coli* strains with varying plaque morphology (S1 Fig in S2 File). This suggests presence of several different phage types and confirms viability of the isolated viral populations (S1 Fig in S2 File) [36]. Temperature variation had no observed effect on plaque formation in our assays.

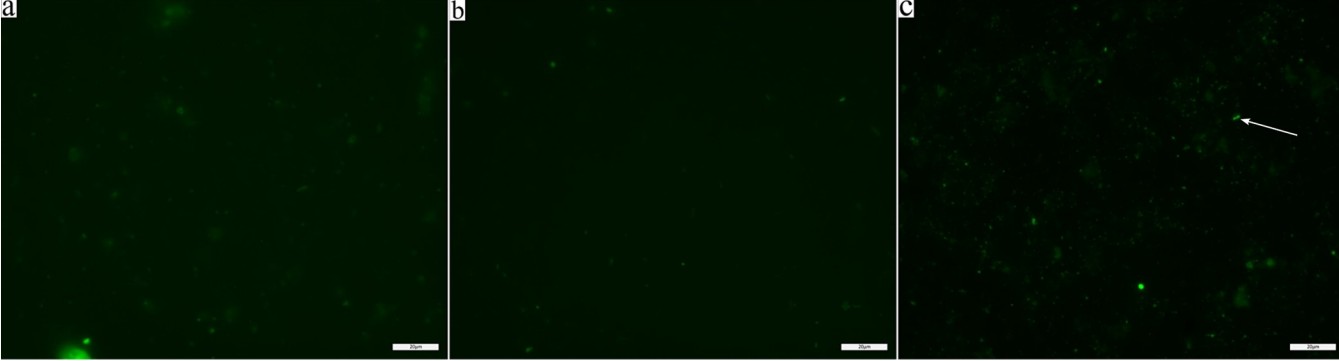

**Fig 1. Phage probes are specific to the targeted phages.** Phage signals are highly specific to the targeted groups. All images are recorded in the FITC filter for phage signals. a) Smear of faecal sample F12 stained with DAPI dye only. Auto-fluorescent debris is vaguely visible. No DNA probes are hybridised to this sample. b) Smear of faecal sample F12 stained with DAPI dye, EUB338-Cy3, nonEUB338-Cy5, and phage probe group E amplified with Alexa Fluor 488 tyramides. Faecal sample F12 does not contain the target sequence for phage probe group E (Table 1), therefore no phage signal is visible and only debris is visible. c) Faecal sample F49 stained with DAPI dye, EUB338-Cy3, nonEUB338-Cy5, and probe group E amplified with Alexa Fluor 488 tyramides. Faecal sample F49 does contain the target phage genome for phage probe group E. Phage signals are clearly visible as single dots of intense signal and intense signals in the shape of single cells (white arrow).

## Primer design and probe synthesis

A DNA target region was selected for each phage for phageFISH detection. 34 primer pairs were designed to amplify several 300bp fragments of the targeted regions (S3 Table in S2 File), 15 of which were successful at amplifying the targets (Table 1). The probes were synthesised from the amplified regions where the primer pairs were successful and for each of the five phages. Seven probes belong to Phage E. For Phage C only a single probe could be synthesised, Phage A and D two probes were synthesised, and for Phage B three probes were obtained (Table 1).

## PCR- and next generation sequencing-based detection of phage across faecal samples

The successful primer pairs were used to confirm the presence of the targeted phage genomes with PCR amplification of the specific phage regions in each faecal sample (Table 1). Each target phage was detected in the samples from which they were originally assembled in (*e.g.*, primer pairs A5 and A6 successfully amplified a 300bp sequences from faecal samples F12 and F33, Table 1). The target phages were also detected across all studied faecal samples to map the presence of each target phage in the individual faecal samples. This was also used to set up negative controls for probe hybridisation (Fig 1). Interestingly, Phage A appears to be present in almost all the tested samples except F67 (from which Phage C was assembled, Table 1) and F59 (from which Phage E was assembled–Table 1), whereas Phage E was found less frequently.

From our next generation sequencing of the viral populations that were isolated from faeces, 10 large-sized phage genomes were *DeNovo* assembled (S1 Table in S2 File). Their sizes ranged between 111 and 336 kb. This suggests that our viral isolations contained larger sized phages directly from the faecal samples.

## PhageFISH set-up and visualisation

Auto-fluorescence in the 475-650nm emission spectrum (FITC filter) is common in faecal samples [37] and was evaluated by comparing one faecal sample (F12) stained with DAPI only (Fig 1A) to samples probed with either target or non-target probes (Fig 1B and 1C). Probe group E was used for the comparison on sample F12 (no Phage E targets, Fig 1B and Table 1) and sample F49 (Phage E target) (Fig 1C and Table 1). Auto-fluorescent debris is detected in the FITC filter spectrum image of the non-probed faecal sample as dim green signals (Fig 1A and 1B). Phage signals, on the other hand, are more distinguishable and defined as described by Allers et al. (as individual intense dots or as intense signals in the shape of bacterial cells) (Fig 1C).

Bacterial cells were probed with cyanine labelled probes (Cy3 and Cy5) instead of HRP-labelled probes amplified by CARD as suggested by Allers et al. This shortened the protocol while still allowing for visible signals from the bacterial probes. Signals produced by cyanine-labelled probes are not affected by the phage probe hybridisation procedure and the nonE-UB338-Cy5 probe could be used as a negative control (Material and methods).

## PhageFISH

All probe groups were able to produce a phage signal in at least one sample (Table 2). To assess the impact of using multiple probes in a sample, the probe groups were compared (S3a-S3e Fig in S2 File), but no obvious difference in signal intensity or abundance was observed between samples that are probed with 1 or 7 probes. This allowed for visual detection of never before isolated phages with unknown hosts directly in the faecal material (Figs 1 and 2 and S4 Fig in

**Table 2.  Phage detection in individual faecal samples by phageFISH in faecal smears, free phages and advanced infection could be detected in all samples.**

| Phage group | Faecal sample | Free phage | Phage infection |
|---|---|---|---|
| A | F12 | Yes | Yes |
| A | F33 | Yes | Yes |
| B | F42 | Yes | Yes |
| B | F71 | Yes | Yes |
| C | F67 | Yes | Yes |
| C | F78 | Yes | Yes |
| D | F95 | Yes | Yes |
| D | F101 | Yes | Yes |
| E | F49 | Yes | Yes |
| E | F59 | Yes | Yes |

S2 File). Phage detection quality appears highly dependent on the faecal sample and the cell distribution in the faecal smears as witnessed by visual inspection of approximately 140 smears.

Bacterial 16S rRNA probe EUB338-Cy3 was hybridised to bacterial cells to study the co-localisation between phage signals and bacterial signals. Co-localising signals indicate phage adsorption and/or infection of the targeted organisms, and further support successful hybridisation to phages. For Phage D, phage signals do not appear to co-localise well with the bacterial signal (Fig 2A and 2B and S4e-S4h Fig in S2 File). Using DAPI dye to visualise all DNA (including fungi, protozoa, mammalian cells) showed better co-localisation (Fig 2C and 2D). However, using DAPI dye also shows a tendency for phage particles to co-localise and have high affinity to the sample debris (Fig 2C and 2D). Phage-bacteria co-signalling is illustrated in detail from one of our samples (faecal sample F49 –Fig 3A and 3B). The blue bacterial 16S rRNA probe signal co-localises with the green phage signal at different loci of the bacterial cell in one image (Fig 2A and 2B). Other examples of co-localisation observations are marked in the images of all samples available in (S4 Fig in S2 File).

## Discussion

Studying novel phages and phage types that are discovered with the rapidly increasing metagenomic research is paramount to our understanding of the ecosystems we investigate. Traditional methods of phage isolation have proven insufficient for isolation of many types of phages [11,25] and new methods reducing this bias are necessary. Here, we attempted to detect five jumbophage *ex vivo* directly in faeces, using adapted fluorescence *in situ* hybridisation method (phageFISH). Hybridisation probes were derived from the five jumbophage genomes there previously described *in silico* [9]. This allowed visualisation of phage-host interaction in a culture-independent fashion for all those phage genomes.

In a previous study where geneFISH, the basis of phageFISH, was used, an approx. 40% detection efficiency for single copy genes using a single probe was estimated [38]. Based on this, Allers et al. argue that single copy targets (such as phage genomes) are unlikely to produce signals strong enough for detection of single free phages and show that detection efficiency increased with each added probe using long probes and CARD signal amplification [34]. Here however, we found it possible to detect free phages (Phage C) using a single 300bp probe (probe C5 in Table 1, S3c Fig in S2 File). This could be due to the variation in the targeted phages between this study and that of Allers et al. The large genomes of the target phages here may behave differently than the small, well-studied phages by Allers et al. [34]. A tendency for

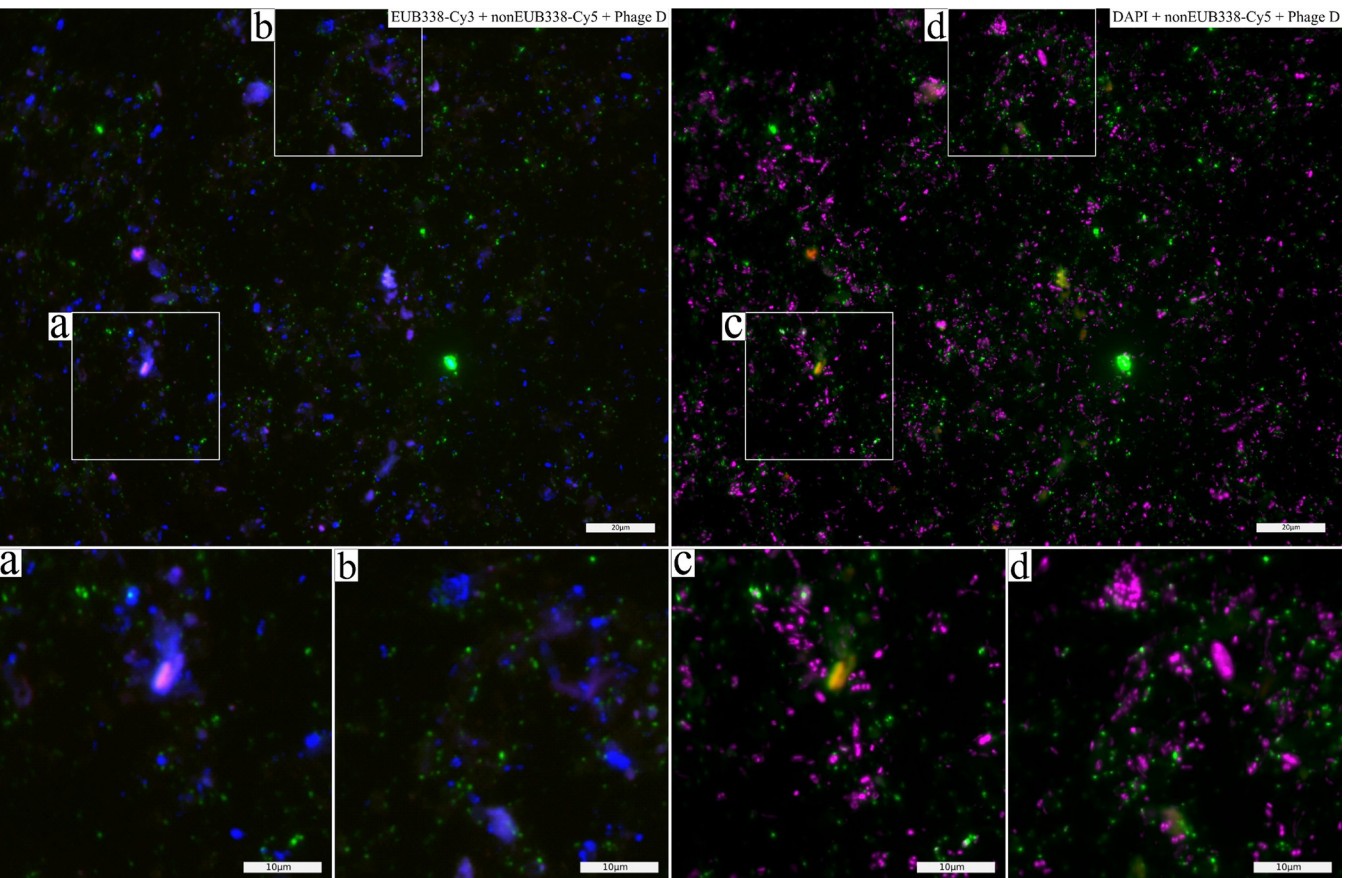

**Fig 2. Co-localisation of phage signals to EUB338-Cy3 and DAPI signals.** Microscopic image of faecal smear (sample F95) hybridised with bacterial probe EUB338-Cy3 (blue, only shown in a and b), and phage group D probes (signal simplified—green), and non-specific probe nonEUB338-Cy5 (negative control—red). All nucleic acids stained with DAPI (magenta, only shown in c and d). Co-localisation with phage signals is better observed with DAPI staining than with the bacterial probe EUB338-Cy3. Scale bars indicate 20µm and 10µm respectively.

phages to adhere to debris in the sample may also affect individual probe detection, as signal strength may increase by accumulation (S2 File).

As opposed to the protocol presented by Allers et al. and Barrero-Canosa & Moraru [34,35], cyanine-labelled probes were used for bacterial gene detection instead of HRP-labelled and CARD-amplified probes. Most bacterial species encode multiple copies of the 16S rRNA gene, making the cyanine fluorescent signal strong enough for detection in normal fluorescence microscopy. This was verified by the optimisation of each Cy-labelled probe in cultures before hybridisation in faeces. We tested CARD-amplification of bacterial genes and found great resolution of cells in culture, but the signals were quenched after phage-probe hybridisation (S2 File).

Cy-labelled probes are a faster and cheaper alternative to DIG-labelled or HRP-labelled probes and met the standards necessary in this study. It also allowed us to use the negative control probe nonEUB338-Cy5 at the same time as both bacterial and phage probes. Furthermore, the nonEUB338-Cy5 probe acted as a control for background fluorescence and auto-fluorescence in the faecal smears.

PhageFISH has previously been used to study phage-host infection dynamics [39,40]. Here, we used a universal bacterial probe to visualise bacterial cells interacting with the phages. Co-localisation of bacterial and phage signals were observed in all samples, but were not common

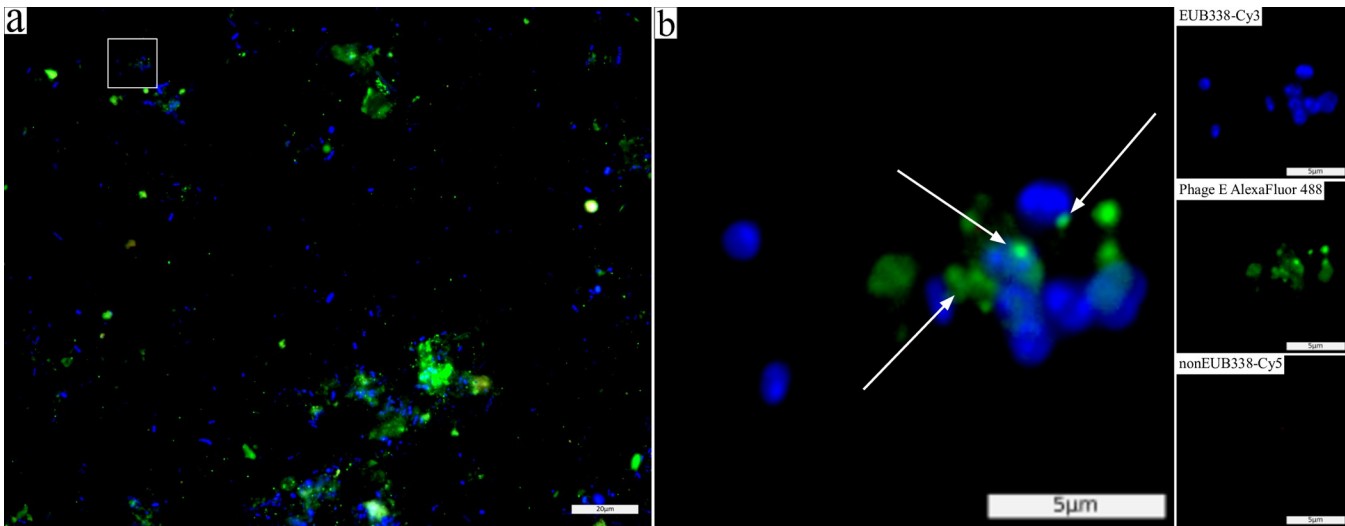

**Fig 3. Co-localisation of phage signals and EUB338-Cy3 signals allows visualisation of phage-host interaction and phage behaviour.** Microscopic image of faecal smear (sample F49) hybridised with phage group E probes (green), universal bacterial probe EUB338-Cy3 (blue), and non-specific probe nonEUB338-Cy5 (red). Various loci of phages were observed within or around the bacterial cells (white arrows in b). For example, phages exist as free-floating phages visible as single signals not co-localised to any bacterial signal, or adsorbed to the outer membrane of the host cell. White box in (a) indicates enlarged area in (b). Panels to the right show separated signals (EUB338-Cy3, Phage probe E, and nonEUB338-Cy5) of the enlarged area. Scale bars indicate 20μm (a) and 5μm (b).

(Table 2, Figs 2 and 3 and S4 Fig in S2 File). This could be explained by the targeted phages and their hosts not making up a large proportion of the sample as many bacteria and their phages are abundant in faecal samples [41]. This also explains the large proportion of free phages (phage signals did not co-localise with bacterial signal) in the samples. As conditions change along the GI tract, phage adsorption to other cells may be inhibited or reduced, causing a large proportion free phages [42–44].

The eubacterial probe (EUB338-Cy3) is widely used in fluorescence *in situ* hybridisation experimental setup and was used with all our pig faecal samples. However, EUB338 has reduced affinity to certain bacterial species [45], the majority of which are not typically associated with animal faeces. The untargeted phylum *Verrucomicrobia*, however, may be present in gut microbiomes and represented by the *Akkermansia* genus [46]. As the relative abundance of *Verrucomicrobia* members in pig faeces is relatively low [42,47], it is unlikely that the detection of free phages with a single probe can be explained as infection of an unstained bacterial cell. This is further supported by the observations of advanced infection in bacterial cells in multiple samples (S4 Fig in S2 File).

As the numbers of phage genomes identified *in silico* become more abundant, (*e.g.*, CrAss phages [25–28]) it is necessary to find new methods of investigation. Isolation of organisms from *in silico* to *in vitro* is not a straightforward process, as a large percentage of sequenced bacteria cannot be cultured *in vitro* increasing the risk of the phage of interest being unculturable too.

The prospect of isolating the phage is complicated by the size of the genome. Relatively few phage genomes of similar size have been described and little is known of the ways in which the increased genomic real estate may influence the function of the phage and the isolation parameters necessary to accommodate the large jumbophage genomes [29,48]. Lak phage, for example, was attempted isolated with no special consideration of phage size, leaving little chance of isolating it in culture [30].

Similarly here, all five phage genomes are based solely on metagenomic assemblies from pig gut microbiomes and are larger than the average phage [48]. PhageFISH and other fluorescent

techniques offer a method of observation in a culture independent fashion. Without any information about potential hosts, the phages were observed directly in their natural environment to achieve unbiased visualisation of the unknown phages. To our knowledge, this is the first example of phage-host detection for a non-isolated phage in faeces with no known host.

In conclusion, the adapted phageFISH method presented here allows detection of jumbo-phages discovered *in silico* in their natural environment regardless of their size, life cycle, and host organism. Further studies combined with more specific and family or species-specific bacterial probes, may allow potential phage hosts identification irrespective of the bacterial host ability to be cultivated. As phages discovered *in silico* have proven difficult to isolate *in vitro*, development of such methods is important to facilitate biological validation of novel phages, as well as characterisation in a culture-independent manor.

## Supporting information

**S1 File. Step-by-step protocol, also available on protocols.io.**
(PDF)

**S2 File. Contains all the supporting text (supporting material and methods), tables (S1 Table–S5 Table) and figures (S1 Fig–S6 Fig).**
(DOCX)

## Acknowledgments

The authors thank Terje Svingen, DTU FOOD, for the help with the fluorescence microscopy. The authors would also like to thank two anonymous reviewers for the their valuable comments and suggestions.

## Author Contributions

**Conceptualization:** Line Jensen Ostenfeld, Patrick Munk, Frank M. Aarestrup, Saria Otani.

**Data curation:** Line Jensen Ostenfeld, Frank M. Aarestrup, Saria Otani.

**Formal analysis:** Line Jensen Ostenfeld, Frank M. Aarestrup, Saria Otani.

**Funding acquisition:** Frank M. Aarestrup.

**Investigation:** Line Jensen Ostenfeld, Patrick Munk, Frank M. Aarestrup, Saria Otani.

**Methodology:** Line Jensen Ostenfeld, Patrick Munk, Saria Otani.

**Project administration:** Line Jensen Ostenfeld, Frank M. Aarestrup, Saria Otani.

**Resources:** Frank M. Aarestrup.

**Software:** Line Jensen Ostenfeld, Patrick Munk.

**Supervision:** Frank M. Aarestrup, Saria Otani.

**Validation:** Line Jensen Ostenfeld, Patrick Munk, Frank M. Aarestrup, Saria Otani.

**Visualization:** Line Jensen Ostenfeld, Frank M. Aarestrup, Saria Otani.

**Writing – original draft:** Line Jensen Ostenfeld, Saria Otani.

**Writing – review & editing:** Line Jensen Ostenfeld, Patrick Munk, Frank M. Aarestrup, Saria Otani.

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
