## [Decision Letter · Decision Letter 0]

12 Jan 2023

PONE-D-22-31700Detection of specific uncultured bacteriophages by fluorescence in situ hybridisation in pig microbiomePLOS ONE

Dear Dr. Otani,

Thank you for submitting your manuscript to PLOS ONE. After careful consideration, we feel that it has merit but does not fully meet PLOS ONE’s publication criteria as it currently stands. Therefore, we invite you to submit a revised version of the manuscript that addresses the points raised during the review process. Please carefully consider all comments provided by the expert reviewer.

We look forward to receiving your revised manuscript.

Kind regards,

Ulrich Nübel

Academic Editor

PLOS ONE

Journal Requirements:

2. We note you have not yet provided a protocols.io PDF version of your protocol and/or a protocols.io DOI. When you submit your revision, please provide a PDF version of your protocol as generated by protocols.io (the file will have the protocols.io logo in the upper right corner of the first page) as a Supporting Information file. The filename should be S1_file.pdf, and you should enter “S1 File” into the Description field. Any additional protocols should be numbered S2, S3, and so on. Please also follow the instructions for Supporting Information captions [https://journals.plos.org/plosone/s/supporting-information#loc-captions]. The title in the caption should read: “Step-by-step protocol, also available on protocols.io.”

Please assign your protocol a protocols.io DOI, if you have not already done so, and include the following line in the Materials and Methods section of your manuscript: “The protocol described in this peer-reviewed article is published on protocols.io (https://dx.doi.org/10.17504/protocols.io.[...]) and is included for printing purposes as S1 File.” You should also supply the DOI in the Protocols.io DOI field of the submission form when you submit your revision.

If you have not yet uploaded your protocol to protocols.io, you are invited to use the platform’s protocol entry service [https://www.protocols.io/we-enter-protocols] for doing so, at no charge. Through this service, the team at protocols.io will enter your protocol for you and format it in a way that takes advantage of the platform’s features. When submitting your protocol to the protocol entry service please include the customer code PLOS2022 in the Note field and indicate that your protocol is associated with a PLOS ONE Lab Protocol Submission. You should also include the title and manuscript number of your PLOS ONE submission.

"NO authors have competing interests."

Reviewers' comments:

Reviewer's Responses to Questions

**Comments to the Author**

1. Does the manuscript report a protocol which is of utility to the research community and adds value to the published literature?

Reviewer #1: Yes

2. Has the protocol been described in sufficient detail?

To answer this question, please click the link to protocols.io in the Materials and Methods section of the manuscript (if a link has been provided) or consult the step-by-step protocol in the Supporting Information files.

The step-by-step protocol should contain sufficient detail for another researcher to be able to reproduce all experiments and analyses.

Reviewer #1: Partly

3. Does the protocol describe a validated method?

Reviewer #1: Yes

4. If the manuscript contains new data, have the authors made this data fully available?

Reviewer #1: N/A

**5. Is the article presented in an intelligible fashion and written in standard English?**

Reviewer #1: Yes

6. Review Comments to the Author

Reviewer #1: General comments:

Ostenfeld et al presented certain novelty in the visualization of jumbophages from metagenome assembled genome, and made certain degree of improvements on the original FISH protocol for the special sample type. However, the whole structure is distractive by some unnecessary information, and the lay out also could be improved by moderate revision:

1. The phage plaque assay (from line 112 - 134) on E. coli cannot significantly contribute to the theme, also kinds of misleading as readers might be confused with the jumbophage’s host is E. coli. I would thus suggest remove this part to avoid the confusion.

2. The sequenced phage contigs (Table S1) didn’t give much information here, only the contig length, is there any host-prediction work was done? According to the contig size, more details should be obtained by additional annotation. Besides, when phage contigs were assembled from 2 different samples, then based on which information that these two contigs were considered to be one? And when designing primers, which contig were used for each phage?

3. In Fig 3, the phage-host relation’s part is less convincing, as the phage clouds around bacteria, but no signal could be observed directly inside a bacterial cell, which is quite abnormal in other FISH publications. Some figure in supplementary material shows better results, may be worth to consider switch the position.

Detailed suggestions:

Also, some small points that could be improved:

1. Line 153: what software used for primer design? Based on what parameters?

2. No signal could be observed for all the nonEUB338-Cy5 probe, in Fig 1 or Fig S4. As I could understand, this is a negative control probe, target on nothing. So, the description in line 189 about the overlapping signal with background is confusing, adding some arrows might be helpful.

7. PLOS authors have the option to publish the peer review history of their article (what does this mean?). If published, this will include your full peer review and any attached files.

Reviewer #1: No

---

## [Author Response · Author response to Decision Letter 0]

9 Feb 2023

Review Comments to the Author

Reviewer #1: General comments:

Ostenfeld et al presented certain novelty in the visualization of jumbophages from metagenome assembled genome, and made certain degree of improvements on the original FISH protocol for the special sample type. However, the whole structure is distractive by some unnecessary information, and the lay out also could be improved by moderate revision:

1. The phage plaque assay (from line 112 - 134) on E. coli cannot significantly contribute to the theme, also kinds of misleading as readers might be confused with the jumbophage’s host is E. coli. I would thus suggest remove this part to avoid the confusion.

“We thank the reviewer for their comment. We agree that this part could be confusing and should be improved. We have now adjusted the text and reduced it to briefly mention the E. coli assays only amongst other tested hosts (Lines: 114-138 – numbers are from the track change version). We believe that our plaque assays with E. coli as a host provide a valid evaluation step that jumbophages can be in vitro detected with plaque assay from those faecal samples. An in vitro detection of jumbophages adds value to the manuscript. However, we do not mention in the text that E. coli is the host to all jumbophages, we only mention that E. coli is potential host to those jumbophages that are isolated from the E. coli plaque assays as a validation step. This was also mentioned in lines 213-220 to avoid confusion.”

2. The sequenced phage contigs (Table S1) didn’t give much information here, only the contig length, is there any host-prediction work was done? According to the contig size, more details should be obtained by additional annotation. Besides, when phage contigs were assembled from 2 different samples, then based on which information that these two contigs were considered to be one? And when designing primers, which contig were used for each phage?

“We thank the reviewer for this comment. The purpose of providing the mentioned phage contigs in Table S1 is solely to show that those phages are of large size (as the reviewer mention in their comment). We refrain from carrying any further analyses on such sequencing output to leave those for an in silico-based studies (such as Al-Shayeb et al. that we used their published phage genomes to design our primers as mentioned in the text). The targeted genomes previously published were isolated from the Danish pig farms in questions. While predictions of bacteriophage hosts from in silico data can be complex, there are several online tools that are currently being developed to serve such purpose (PHERI, HostPhinder, Host Taxon Predicter, WIsH, MARVEL, Seeker, Phirbo, VirHostMatcher and others) using metagenomic sequencing from Illumina platform (which is the common practice). We therefore refrain from carrying out any of those predictions from our contig information in table S1, as the main purpose of the study is to develop an in vitro visualisation method to microscopically detect jumbophages (and in vitro detect them hence the E. coli host culturing in lines 213-220). 

Regarding the phage contigs and how they were handled, we agree it was not clear from the manuscript and have now updated it accordingly. The phage complete and draft genome scaffolds were derived from the Al-Shayeb study, where analysis on whether these hundreds of large circular DNA entities were phage-like, plasmid-like and shared similarity to known sequences. The genomes used for primer design here were therefore much larger and often circular (200Kbp+ jumbo-phage genomes). Pairs of these previously published, high-identity genomes derived from different Danish pig farms were then aligned using progressive Mauve and inspected for especially conserved regions. Such high-identity genomic regions (>300bp) were visually identified, extracted and used as input for the version of Primer3 integrated in Geneious software. We looked for high-scoring primer pairs (20-30bp with a 25bp optimum), with 300bp product sizes and a Tm around 60C (57-60C). Several such pairs were identified for each target genome and the PCR probe products were then searched against NCBI nt using BlastN to ensure specificity. The text in the manuscript was adjusted accordingly (Lines 157-164).”

3. In Fig 3, the phage-host relation’s part is less convincing, as the phage clouds around bacteria, but no signal could be observed directly inside a bacterial cell, which is quite abnormal in other FISH publications. Some figure in supplementary material shows better results, may be worth to consider switch the position.

“We thank the reviewer for this suggestion. We chose this sample (F49) and its phageFISH images to be in the main text as it shows potential signals for various stages of phages: within the bacterial cells as an advanced stage of infection (red arrow in Fig S4 which is the original image for Fig 3 – and the top left arrow in the current Fig 3) and potentially early stages of infections (yellow arrow in Fig S4 which is the original image for Fig 3 – and the top right arrow in the current Fig 3). This also in agreement with the findings from Allers et al., 2013 showing various stages of phage infection with phageFISH, where phages were not only within the cells, but also surround the bacterial cells.”

Detailed suggestions:

Also, some small points that could be improved:

1. Line 153: what software used for primer design? Based on what parameters?

“The primers were designed using Primer3 v.2.3.7 in Geneious with mostly default parameters, other thans: primer length: 20-30, optimal primer length: 25, Tm: 57-63, product length: 300 bp. The text was adjusted accordingly in lines: 157-164. Please see the answer to the point above for more details on primer design and the sequences used.”

2. No signal could be observed for all the nonEUB338-Cy5 probe, in Fig 1 or Fig S4. As I could understand, this is a negative control probe, target on nothing. So, the description in line 189 about the overlapping signal with background is confusing, adding some arrows might be helpful.

“We thank the reviewer for this, we agree. We have now added an arrow to Fig 1 to indicate the targeted signal and adjusted the figure legend. Same to S5 and S6 Figs. We have also edited the text in that part of the main manuscript to make it clearer in lines: 202-209” 

References: 

Al-Shayeb B, Sachdeva R, Chen LX, et al. Clades of huge phages from across Earth’s ecosystems. Nature. 2020;578:425-431. doi:10.1038/s41586-020-2007-4

Allers E, Moraru C, Duhaime MB, et al. Single-cell and population level viral infection dynamics revealed by phageFISH, a method to visualize intracellular and free viruses. Environ Microbiol. 2013;15(8):2306-2318. doi:10.1111/1462-2920.12100

---

## [Editor Report · Decision Letter 1]

14 Mar 2023

Detection of specific uncultured bacteriophages by fluorescence in situ hybridisation in pig microbiome

PONE-D-22-31700R1

Dear Dr. Otani,

We’re pleased to inform you that your manuscript has been judged scientifically suitable for publication and will be formally accepted for publication once it meets all outstanding technical requirements.

Kind regards,

Ulrich Nübel

Academic Editor

PLOS ONE
---

## [Editor Report · Acceptance letter]

20 Mar 2023

PONE-D-22-31700R1 

Detection of specific uncultured bacteriophages by fluorescence *in situ* hybridisation in pig microbiome 

Dear Dr. Otani:

I'm pleased to inform you that your manuscript has been deemed suitable for publication in PLOS ONE. Congratulations! Your manuscript is now with our production department. 

Kind regards, 

on behalf of

Dr. Ulrich Nübel 

Academic Editor

PLOS ONE